# Integrating Spatial and Temporal Approaches for Explaining Bicycle Crashes in High-Risk Areas in Antwerp (Belgium)

**Hwachyi Wang [1,2,3,\*], S. K. Jason Chang [1], Hans De Backer [3], Dirk Lauwers [2,3] and Philippe De Maeyer [4]**

[1] Department of Civil Engineering, National Taiwan University, No. 1, Sec. 4, Roosevelt Road, Taipei 10617, Taiwan
[2] Center for Mobility and Spatial Planning, Ghent University, Sint-Pietersnieuwstraat 41 B2, B-9000 Gent, Belgium
[3] Department of Civil Engineering, Technologiepark Zwijnaarde 904, Ghent University, B-9052 Zwijnaarde, Belgium
[4] Department of Geography, Krijgslaan 281 S8, Ghent University, B-9000 Gent, Belgium
[\*] Correspondence: hwachyi.wang@UGent.be

**Abstract:** The majority of bicycle crash studies aim at determining risk factors and estimating crash risks by employing statistics. Accordingly, the goal of this paper is to evaluate bicycle–motor vehicle crashes by using spatial and temporal approaches to statistical data. The spatial approach (a weighted kernel density estimation approach) preliminarily estimates crash risks at the macro level, thereby avoiding the expensive work of collecting traffic counts; meanwhile, the temporal approach (negative binomial regression approach) focuses on crash data that occurred on urban arterials and includes traffic exposure at the micro level. The crash risk and risk factors of arterial roads associated with bicycle facilities and road environments were assessed using a database built from field surveys and five government agencies. This study analysed 4120 geocoded bicycle crashes in the city of Antwerp (CA, Belgium). The data sets covered five years (2014 to 2018), including all bicycle–motorized vehicle (BMV) crashes from police reports. Urban arterials were highlighted as high-risk areas through the spatial approach. This was as expected given that, due to heavy traffic and limited road space, bicycle facilities on arterial roads face many design problems. Through spatial and temporal approaches, the environmental characteristics of bicycle crashes on arterial roads were analysed at the micro level. Finally, this paper provides an insight that can be used by both the geography and transport fields to improve cycling safety on urban arterial roads.

**Keywords:** urban arterial roads; geographic information system; bicycle–motorized vehicle (BMV) crashes; spatial; weighted kernel density estimation; temporal; negative binomial; crash severity index

## 1. Introduction

As a key component of sustainable transportation systems, cycling has been actively promoted in cities throughout the world [1,2]; however, bicycle-related crashes have been associated with increasing numbers of fatalities and injuries [3–6] and the risk of crashes prevents people from using bicycles [7]. Compared with driving, cyclists have a higher probability of injuries in traffic accidents [8]. Unfortunately, bicycle crash risks are unclear [7] because current risk estimates mainly depend on general exposure (such as population or census data) [9,10] or insufficient exposure (such as traffic exposure or bicycle exposure) [11]. Despite the fact that bicycle crashes tend to cluster within a spatial

area [12,13], and are likely to be over-dispersed during a time period, the majority of risk studies provide no explanation for these spatio-temporal aspects of bicycle crashes.

### 1.1. Research Gap

Bicycle crashes are mainly studied from the perspectives of transportation and geography. From the transportation perspective, most studies neglect the regional impact of bicycle safety on the macro-scale level. This holds especially true for crash analyses of specific locations, e.g., intersections or other crash-prone locations on the micro-scale level. Meanwhile, from a geographical perspective, many studies ignore the locational influence of safety attributes on the micro-scale level. This especially applies to crash analyses of geographic areas, such as spatial autocorrelation or many different kinds of cluster analyses on the macro-scale (regional) level. Moreover, from an epidemiological point of view, bicycle crashes are complex spatio-temporal phenomena with many contributing risk factors (e.g., weather conditions, motorized and non-motorized traffic volumes, road facilities, road traffic controls and driving behaviours) varying over space and time. Therefore, significant gaps are present in current bicycle crash studies.

### 1.2. Research Goals

In order to determine future measures to improve cycling safety, this paper aims to:

1. Reveal the spatial and temporal risk patterns of bicycle crashes (where? and when?) from a regional level to a locational level (a macro scale to a micro scale). To this end, a two-stage workflow (spatial and temporal approaches) is created for exploring bicycle crashes. Through the spatial approach, urban arterials are determined to have the highest bicycle–motorized vehicle BMV crash densities (see Figure 6a and Figure 7).
2. Explain how arterial infrastructure affects bicycle crashes in the city of Antwerp (CA, Belgium) by examining possible risk factors.

## 2. Background of Spatio-Temporal Approaches

### 2.1. Bicycle–Motorized Vehicle (BMV) Crash Studies

Bicycle–motorized vehicle crashes raised concerns in the 1970s when the annual fatality rate among children surpassed 500 in the Netherlands [14]. Since the 1990s, much research has focused on road crashes related to bicycles. Road crashes arise from the interaction between human factors (such as driving behaviours), environmental factors (such as traffic exposure, road facilities), and vehicle-related factors (e.g., driving speed) [15–17]. These factors account for 57%, 34%, and 13% of all bicycle crashes, respectively (Figure 1a) [18]. However, recent reviews on BMV crashes have identified that these factors account for 59.3%, 57.6%, and 15.3%, respectively [19]. As can be seen from Figure 1b, the influence of environmental factors in crashes involving bicycles and motorised vehicles may be comparable to that of human factors.

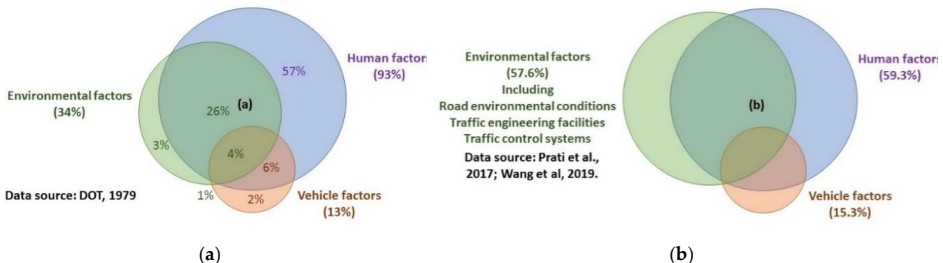

**Figure 1.** (**a**) Road crashes are mainly influenced by the interaction of human, vehicle and environmental (road) factors [18]. (**b**) Recent studies show that BMV crashes are mainly caused by the interaction of human, vehicle and environmental factors, where environmental factors include road environments, traffic engineering and road traffic controls [20,21].

A number of previous studies have explored the relationship between bicycle crashes and the physical environment [8,22,23]. Additionally, despite limitations due to issues of privacy, a few studies have explored the influences of human and vehicle factors on bicycle collisions [24–26].

*2.2. BMV Crash Patterns*

BMV crash patterns are fundamentally space- and time-related because the occurrence of a crash directly involves its specific location and time. Since bicycle crashes are mainly assessed through the perspectives of geography (which is related to spatial methodologies) and transportation (which is related to temporal methodologies), it is essential to review the current literature relating to these approaches.

2.2.1. Spatial Patterns of BMV Crashes

Many risk evaluations have focused on spatial crash analysis. Bíl [27] and Chen [28] studied the crash risk of road sections, and Hels and Orozova-Bekkevold [29] and Harris et al. [30] investigated the risk-influencing factors of road roundabouts and intersections, respectively. Compared with road sections, the majority of studies found an increased possibility of BMV crashes at intersections [4,28,31–33]; however, the likelihood of severe injuries from BMV crashes on the road sections is more likely to increase [4]. In addition, Vandenbulcke et al. [34] indicated that distinguishing cities (built-up areas) from rural areas is important because bicycle crashes are closely associated with urban structures [21,35,36].

With regard to the spatial scale, crash risks can be assessed from a regional level (macro scale) [37–42] to a locational level (micro scale) [43,44]. Generally speaking, in the regional (macro) level, BMV risks can be analysed over certain geographic zones in order to understand the effect of environmental factors on accident occurrences, thereby improving traffic safety in the whole region [45]. On the other hand, in the locational (micro) scale, BMV risks can be observed on certain specific road entities (e.g., ramps, curved road sections) in order to determine explanatory factors contributing to accident events and thereby facilitating constructive countermeasures to reduce crashes [20]. However, few BMV crash studies so far have estimated the risk of certain specific locations (e.g., urban arterials, tunnels and bridges) [9]. The spatial approaches proposed in this paper provide generalisability, meaning the same countermeasures may be applied to areas with similar infrastructure characteristics [46].

2.2.2. Temporal Patterns of BMV Crashes

Although less bicycle risk estimations emphasise temporal approaches based on probability modelling due to inadequate time-related counting data (exposure), these approaches focus on the study of contributing factors that affect the frequency of BMV crashes occurring within a certain period [47,48]. A potential alternative approach is to collect exposure data at each location [49]. However, when there are high rates of bicycle collisions, such collected data are increasingly expensive and labour-intensive to obtain [12]. On the other hand, the definition of temporal approaches may refer to only time-related contributing factors (rather than time-related methodologies), where these estimations focus on bicycle risks influenced by traffic exposure, travel time, specific seasons, days, peak hours and road surface conditions related to the weather [20]. Dozza [50] examined cycling risk at hourly, daily, weekly, monthly and seasonal scales and discovered that BMV and single-bicycle crashes had significant risk differences in the dark and during weekend afternoons, peak hours and July. Kim [51] and Yan [52] both indicated that serious bicycle crashes occurred during peak hours, bad weather influencing driver visibility and in the absence of street lights at night. Chen [53] and Prati [21] showed that transient poor road surface conditions (e.g., following rain) may significantly increase the riding risk. Chen and Fuller (2014) [54] found that the probability of a bike crash increased by 2.13 times at night. In addition, early studies indicated that for a given location, rainfall was related to a 70% higher crash risk [55], and it might increase the severity of BMV crashes as well [56].

Note that the temporal analysis of bicycle crashes not only refers to time-related contributing risk factors, but also refers to a BMV crash methodology with exposure counting data (see Section 2.3).

*2.3. BMV Crash Methodology*

2.3.1. Methods without Exposure Counting Data

From a methodological perspective, BMV crash estimates may be separated into two groups: exposure-based and non-exposure-based methods. So far, due to the lack of qualitative traffic counts within a location scale, few bicycle risk estimates have been investigated [9,24,53]. Non-exposure-based methods are viewed as preliminary steps to examine bicycle risks because they explore factors that may cause bicycle crashes with the use of descriptive statistics (such as medical records [57], police and questionnaire data [58]) or general statistics (such as odds ratios [59], logistic models [53,60], census data combined with spatial analyses [61]).

Of these spatial analyses, methods without exposure commonly analyse BMV crashes on a macro scale, such as traffic analysis zones [37,39,41,46], and are implemented in geographic information systems (GIS) spatial modelling, such as kernel density estimations [7,20,62–64]. Since methods with exposure counting data are quite data-intensive, methods without exposure reduce the amount of required data and are important for crash locations with inadequate exposure.

2.3.2. Methods with Exposure Counting Data

Unlike the methods discussed above, exposure-related approaches using related statistical probabilities (such as Bayesian distributions [21], negative binomial distributions [41,43], Poisson distributions [46]), pay more attention to explaining the relationships between influencing factors and bicycle crashes. In addition, because "exposure to risk" is strongly related to average daily bicycle traffic (ADB) and average daily traffic (ADT) [11,65,66], the modelling is labour-intensive and time-consuming. However, exposure-based methods may have a better ability to explain their contributing factors [10,20]. Moreover, by combining these methods with spatial network reference units, such as hotspot analysis [67], cluster analysis [68], or density estimations [67], exposure-based methods may bring more accurate results.

Methods with exposure are strongly related to time because a crash probability model can be developed by controlling bicycle and motorized-vehicle exposure within a specified period to gain better understanding of risk factors [20,47]. However, exposure data are usually unknown [63], the role of exposure can be replaced or explained using (bicycle) lane kilometres [31,69], total number of trips [46], travel behaviour of cycling [49], (bicycle) commuting flows [63] or peak hour flows [70].

*2.4. Risk Factors Associated with Cycling Environments*

Considering the risk impact of cycling environments, the majority of studies have indicated that bicycle crashes are affected by road environments, road traffic controls, and road engineering facilities [8,20]. Particularly, some cycling environments may have an increased risk for bicycle crash frequency but a decreased risk for bicycle crash severity (such as highly-urbanized areas [44,71], peak hours [7], straight roadways [53], signalised intersections). The observed phenomenon may be entirely different, which means frequency risks may be lower but the severity of risks may be higher (e.g., curved road sections [51]). However, in most of the cases, crash frequencies may be consistent with their severity [20]. Overall, previous studies showed that heavy traffic (during peak hours), an increased number of lanes and a higher speed limit may lead to higher crash probabilities. However, up to now, a large number of studies have examined only risk factors related to the crash frequency [71] or severity [21,53]. For example, lower speed limits significantly lower bicycle and pedestrian crashes on urban road sections and junctions [4,59]; the turning movement of bicycles and motorised vehicles and urban road networks are aggravating and mitigating factors for bicycle crashes, respectively [4]; or increased bicycle exposure may reduce the severity of bicycle crashes [36,72,73].

Finally, based on the overall background of bicycle crashes, Section 5 collected 33 potential influencing factors with their detailed descriptions. The study assumed that other factors related to infrastructure (such as primary roads or secondary roads) may affect the risk of having a bicycle crash. Indeed, in the first stage of this study (the spatial approach), urban arterials are identified as high-risk areas, which may be attributed to complex traffic situations in road environments, including amongst others: higher traffic volume, more lanes and wider road widths [7,19,71]. However, prior bicycle crash studies have given little specific information about such risks, how these risks affect BMV crashes, and how they are affected by their BMV crash severities [7,20]. Therefore, the purpose of this research is to explore the patterns of bicycle crashes spatially and temporally, and to reveal how road environments influence bicycle crash risks. A further purpose is to explicate particularities in a case study (urban arterials of city of Antwerp) and reflect on existing BMV risk studies.

## 3. Methodology: Integrating Spatial and Temporal Approaches

For the spatial and temporal analysis of bicycle crashes, a two-stage workflow (Figure 2) integrated geocoded data (such as X and Y coordinates, road sections, road junctions, among others), visual data (such as crash figures, aerial photos), traffic data and infrastructure data into the spatio-temporal analysis (for details, see Section 5). In stage 1, spatial references of analysis combined nine reference units with census data (the population at risk) for analysis. Within each reference unit, BMV crashes with corresponding severity levels were precisely located. Stage 1 included census data because an increased population may contribute to more BMV crashes. Using kernel density estimation as a spatial approach, urban arterials were found to have the highest density of bicycle crashes on a macro scale. The displayed crash patterns were further investigated based on road environments and statistical data by including more information on road networks (e.g., road sections, road junctions, annual updated transportation facilities, etc.). In stage 1, all bicycle crash data were implemented in ArcGIS 10.6.1 (A. 10.6.1; Esri: Redlands, CA, USA, 2019), and in stage 2, only urban arterial crashes were implemented in both ArcGIS 10.6.1 and Limdep 11 (L. 11; Econometric Software, Inc.: Plainview, NY, USA, 2016). Therefore, certain types of bicycle crashes, such as crashes on arterials, can be easier identified and further investigated.

Stage 2 defined the traffic volume data (along with bicycle count data) as temporal references for BMV crashes, and delineated potential influencing factors of arterial crashes. These factors varied over time (such as the timing when a bicycle facility was built) and remained independent through a correlation coefficient test. Negative binomial modelling was used as a temporal approach to estimate the risk of BMV crashes on a microscopic scale. Finally, risk factors associated with bicycle infrastructure were assessed by adopting a maximum likelihood estimation (MLE) and all arterial crash data were put in the Limdep environment.

### 3.1. The Selection of Spatio-Temporal Approaches

3.1.1. Stage 1: Spatial Approach

To detect the spatial pattern of crash events, a weighted kernel density estimation (wKDE) [74,75] served as an initial step in the risk calculation to estimate crash density (Equation (1)). The wKDE placed a surface on each crash point, calculating the distance from a reference position to that point (di); wKDE was generally derived using:

$$D(x,y) = \frac{1}{nh^2} \sum_{i=1}^{n} W_i\, K\!\left( \frac{d_i}{P(u,v)} \right) \tag{1}$$

where the kernel function ($K$) was the intensity of the crash event $i$, influenced by population and distance; $h$ was the search bandwidth (radius); $n$ was the frequency of observed crashes; $D(x,y)$ was the density at the position with x and y coordinates; $W_i$ was the weighted value of the crash event

*i*, which considered the different severity of bicycle crashes [76]. Stage 1 replaced the fixed search bandwidth (*h*) with an adaptive search bandwidth $P(u,v)$ in order to exclude uneven spatial distribution caused by the population at risk. Local population density was presented by the *P* function and was placed at the centre position $(u,v)$ for reference. KDE was first applied to traffic crashes in 2008 [13,77]. The wKDE is the extension of KDE and has the advantage regarding determining the density of crash risks more accurately than KDE [74,75]. Second, such a spatial density approach makes a visual comparison analysis (Figure 7), providing risk class homogeneity for the entire studied area and enabling identification of high-density areas, such as arterial roads in this study.

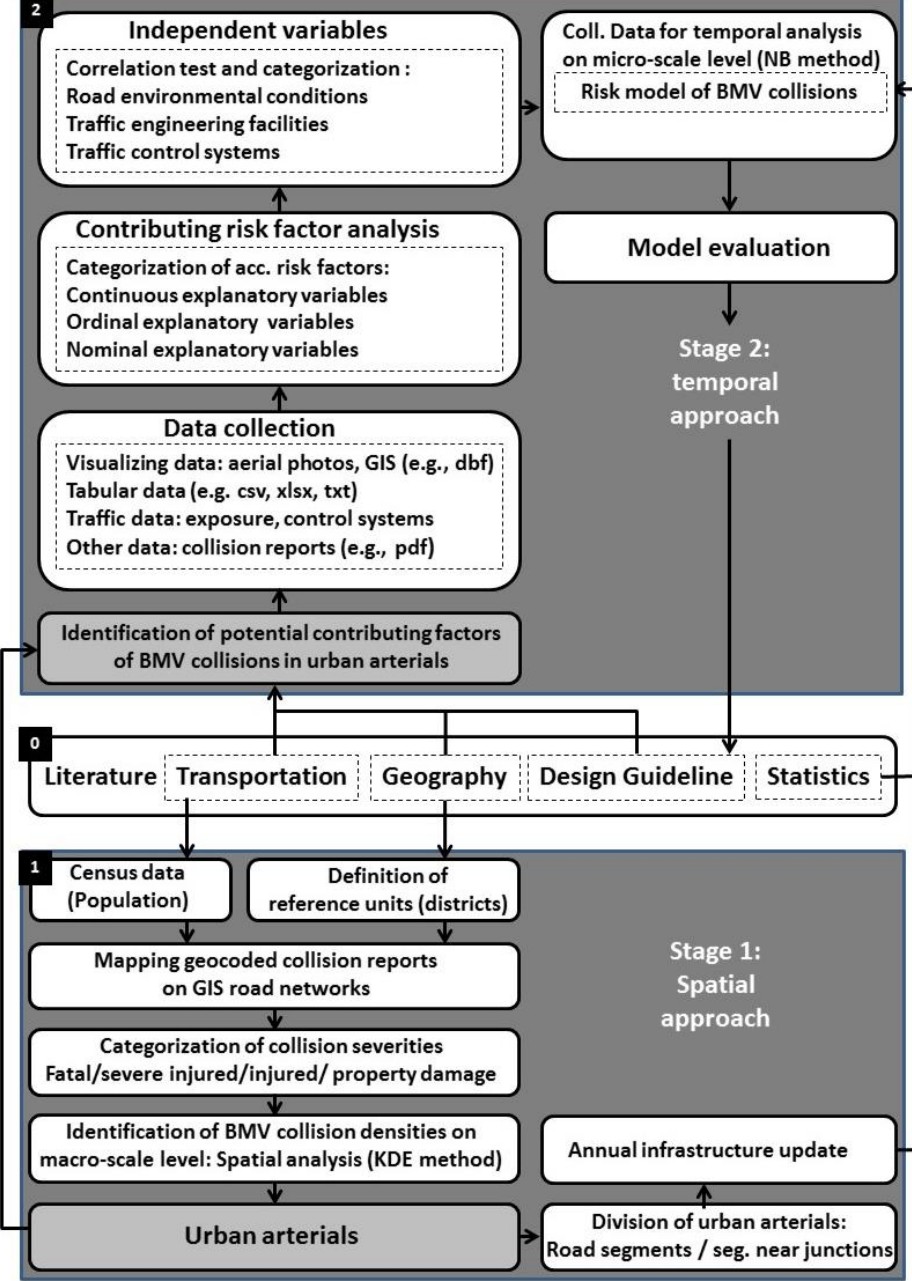

**Figure 2.** A two-stage (spatio-temporal) workflow for bicycle motorised vehicle crashes.

Of note, this study proposes to use planar KDE rather than network KDE [13,78] for density estimations because: (1) By using an adaptive search radius, KDE may easily exclude the uneven spatial distribution caused by the population at risk. (2) KDE (Figure 3a) calculates the density on

an area unit rather than that on a linear unit. However, a crash may be more properly recognised as a spatial point carrying a spread of risk (e.g., crashes caused by a discrepancy in elevation in the mountain areas [20]). (3) With a re-weighting function, wKDE can be adjusted to control for the severity of different crash points (e.g., slightly injured, severely injured, fatal crashes). However, network KDE does not distinguish crashes by injury severity, which means all bicycle crashes are treated the same. (4) Network KDE could possibly underestimate crash density. For example, KDE finds eight bicycle crashes within the search bandwidth, but network KDE only finds three bicycle crashes at the same studied area (Figure 3b) because network KDE does not consider a prolonging distance caused by road curvature or driving manoeuvres.

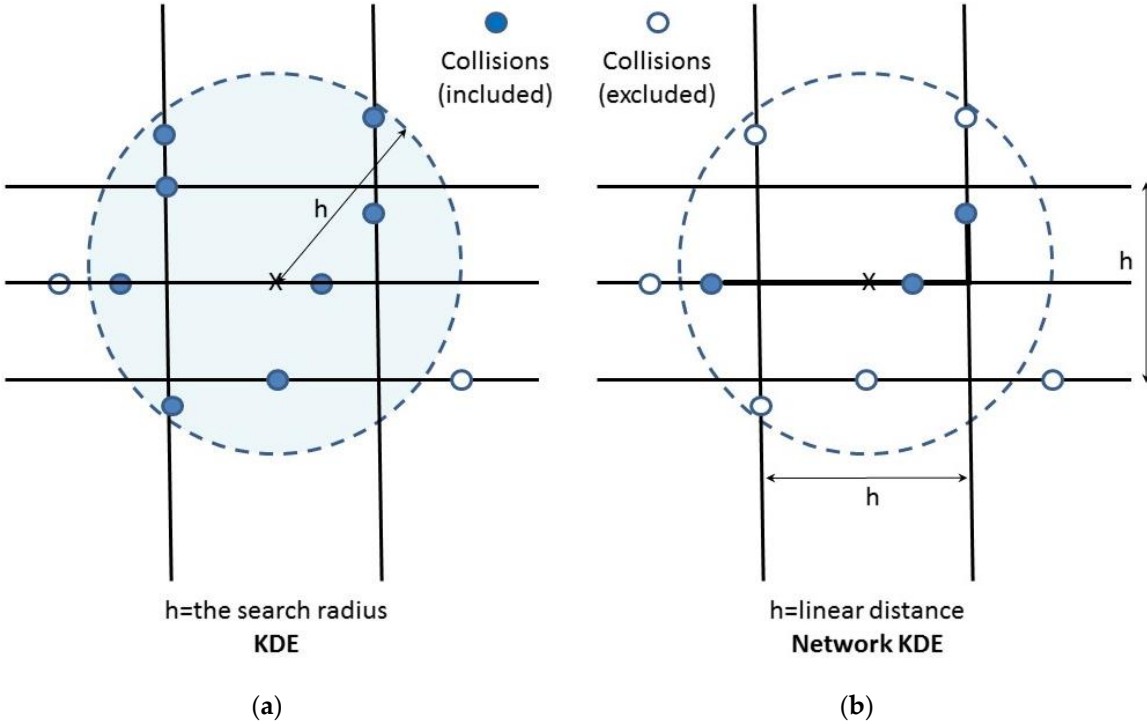

**Figure 3.** The different estimates between (planar) KDE (**a**) and network KDE (**b**) for the same crash points. To estimate the density value, KDE uses the whole two-dimensional space based on Euclidean distances and finds eight crash points within a search radius *h*, whereas network KDE only finds three crash points within the same radius in the network space based on linear distances.

### 3.1.2. Stage 2: Temporal Approach

The temporal approach estimated the crash risks with a suitable risk model. Temporal construction sites or bad road environments may accumulate more BMV crashes. With the help of aggregation by time, the characteristics of BMV crashes can be statistically described and visually detected (stage 2 of Figure 2). A Poisson distribution [47,79] was used for crash probability modelling by controlling motorized vehicle and bicycle exposure within an observed time. This modelling assumed that BMV crashes were random and obeyed a binomial distribution [71] in the observed period. According to this particular distribution, the probability of crashes and the influencing factors related to the probability of crashes can be estimated.

However, comparing the traffic flows ($V_i$) and bicycle risk ($P_i$), $P_i$ had a very small value because the motorized vehicle flows were usually much greater than the frequency of bicycle crashes. Therefore, a Poisson distribution was suitable to explain the binomial distribution of bicycle crashes [80]. The Poisson distribution with random, discrete, and non-negative characteristics was commonly applied to crash estimates; however, this distribution required that its mathematical expectation and variance were equal. In most situations, it was difficult to reach this constraint because the crash data were

over-dispersed. To relax this constraint, inserting an independent error term $\varepsilon_i$ into the Poisson distribution allowed for the variance and mathematical expectation of the Poisson distribution to be unequal [61,81,82].

Moreover, it was assumed that, $e^{\varepsilon_i}$ obeyed a gamma distribution, giving a variance of $\delta$, and $\theta = 1/\delta$, thus a negative binomial (NB) distribution may be more suitable to explicate the prior Poisson distribution of bicycle crashes because it allowed the NB and Poisson distributions' variances to be different and their expected values to remain the same. The negative binomial modelling was given by (the detailed formula derivation can be found in recent research conducted by Wang et al. (2019) [20]):

$$P(n_i|\varepsilon_i) = \frac{\prod(n_i + \theta)}{\prod(n_i + 1)\prod(\theta)}\left(\frac{\theta}{V_iP_i + \theta}\right)^{\theta}\left(\frac{V_iP_i}{V_iP_i + \theta}\right)^{n_i} \tag{2}$$

$$P_i = \frac{F_i}{F_i + e^{-\beta X_i}} \tag{3}$$

$$P(n_i) = \sum_{i=1}^{1415} \frac{\prod(n_i + \theta)}{\prod(n_i + 1)\prod(\theta)}\left(\frac{\theta\left(F_i + e^{-\beta X_i}\right)}{V_iF_i + \theta(F_i + e^{-\beta X_i})}\right)^{\theta}\left(\frac{V_iF_i}{V_iF_i + \theta(F_i + e^{-\beta X_i})}\right)^{n_i} \tag{4}$$

where $i$ was the road junction or section index (see Figure 4a,b), $n_i$ was the number of BMV crashes under specific $V_i$ traffic flows, $V_i$ was the traffic flow of location $i$, $P_i$ was the BMV risk under traffic flow ($V_i$) and $P(n_i|\varepsilon_i)$ was the likelihood of $n_i$ bicycle crashes occurring under the inserted error term $\varepsilon_i$. If $\theta$ was statistically significant, the NB distribution would be used in the BMV crash model. Otherwise, the Poisson distribution would be adopted. Equation (3) shows that the BMV crash risk ($P_i$) was determined by the bicycle flow $F_i$ of location $i$ and a series of influencing risk factors ($X_i$), and $\beta$ was a vector of coefficients of $X_i$. To avoid the possible over-estimation of bicycle risk in stage 1 (Figure 6a), and to reflect the high cycling population at certain locations, $F_i$ was collected by this study according to field surveys on arterial roads. By adding the value of $P_i$ from Equation (3) into Equation (2), the final formula is shown in Equation (4) for the probability $P(n_i)$ of having $n_i$ crashes.

This temporal approach hadd three advantages: (1) The crash risk approached 0 when there were few bicycle flows at location $i$. (2) $\beta$ was made up of a series of coefficients. If the value of $\beta$ was positive, it had an aggravating impact on the crash risk; otherwise, it had a mitigating impact on risk when the value of $\beta$ was negative. (3) Unlike Wang et al. [20], historical crash severity index (CSI) was viewed as a contributing risk factor [20,41] integrated into the temporal modelling (rather than two separated models). Conducting the modelling this way estimated the crash risk on arterial roads because stage 1 had found that the severity of arterial crashes was in line with its crash frequency, which meant that on urban arterials, a location was found with an increased number of crash frequency, accompanied with increased crash injuries and fatalities. Determining and dealing with the road features of these collision-prone positions may greatly lower the risk of crash frequencies, severe injuries and fatalities.

To find the contribution of the crash risk, each influencing risk factor should be independent. Using Pearson's correlation examination, a pair of risk factors with a correlation ≥0.700 would not be together into the model [61,83], only the one with favourable explanatory abilities was included in the temporal modelling [84]. A total of 33 vectors of explanatory variables $X_i$ were later selected for the temporal modelling. These explanatory risk factors ($X_i$) were chosen based on crash types and their environmental features, which may have an aggravating or mitigating influence on the crash risk of urban arterials. Finally, using maximum likelihood estimation (MLE) [47], Equation (5) evaluated the unknown vector of coefficient β:

$$\text{MLE}(\beta, \theta) = \sum_{i=1}^{1415} \frac{\prod(n_i + \theta)}{\prod(n_i + 1)\prod(\theta)}\left(\frac{\theta\left(F_i + e^{-\beta X_i}\right)}{V_iF_i + \theta(F_i + e^{-\beta X_i})}\right)^{\theta}\left(\frac{V_iF_i}{V_iF_i + \theta(F_i + e^{-\beta X_i})}\right)^{n_i} \tag{5}$$

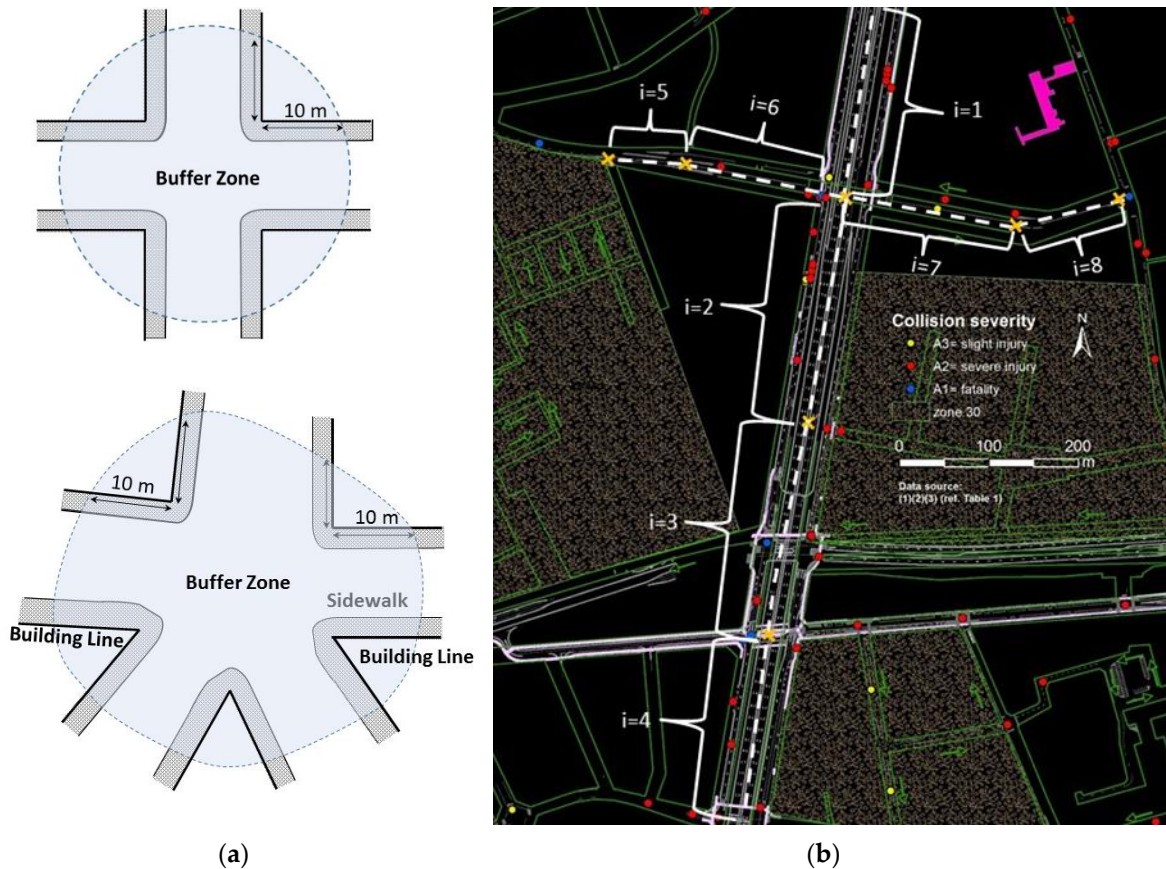

| (**a**) | (**b**) |

**Figure 4.** (**a**) A road junction was defined using 10 m linear buffers from the building line's corner. This definition considered the possible influence caused by traffic signs, traffic signals, and the turning movement of traffic flows. The definition came from the CA Police Department and Department of Transportation (DOT). (**b**) An arterial road was separated into several sections based on each crash event, corresponding to its geometric environments. For example, section index *i* = 1–8 had its own numbers of lanes, section lengths, and other different road features.

## 4. Case Study

The presented case study considered both statistical population and exposure counting data. BMV crashes were spatially and temporally combined. The proposed spatio-temporal approaches were applied to a case study of the city of Antwerp (Belgium) where a five-year BMV crash database was analysed, from 2014 to 2018. Before Section 6 presents the results, Sections 4 and 5 describe the data, studied area, and the detailed information regarding contributing risk factors.

*Study Area*

The city of Antwerp (CA) had approximately 521,000 residents and the population density was approximately 2500 residents per square kilometre. In the CA, the total length of the road network related to bicycle flow was 6269 kilometres. Around 44% of these roads (2750 kilometres) had bicycle infrastructure in the range of shared uni-directional bicycle lanes to separated bi-directional bicycle lanes, where 498 kilometres were exclusive cycle lanes (Figure 5).

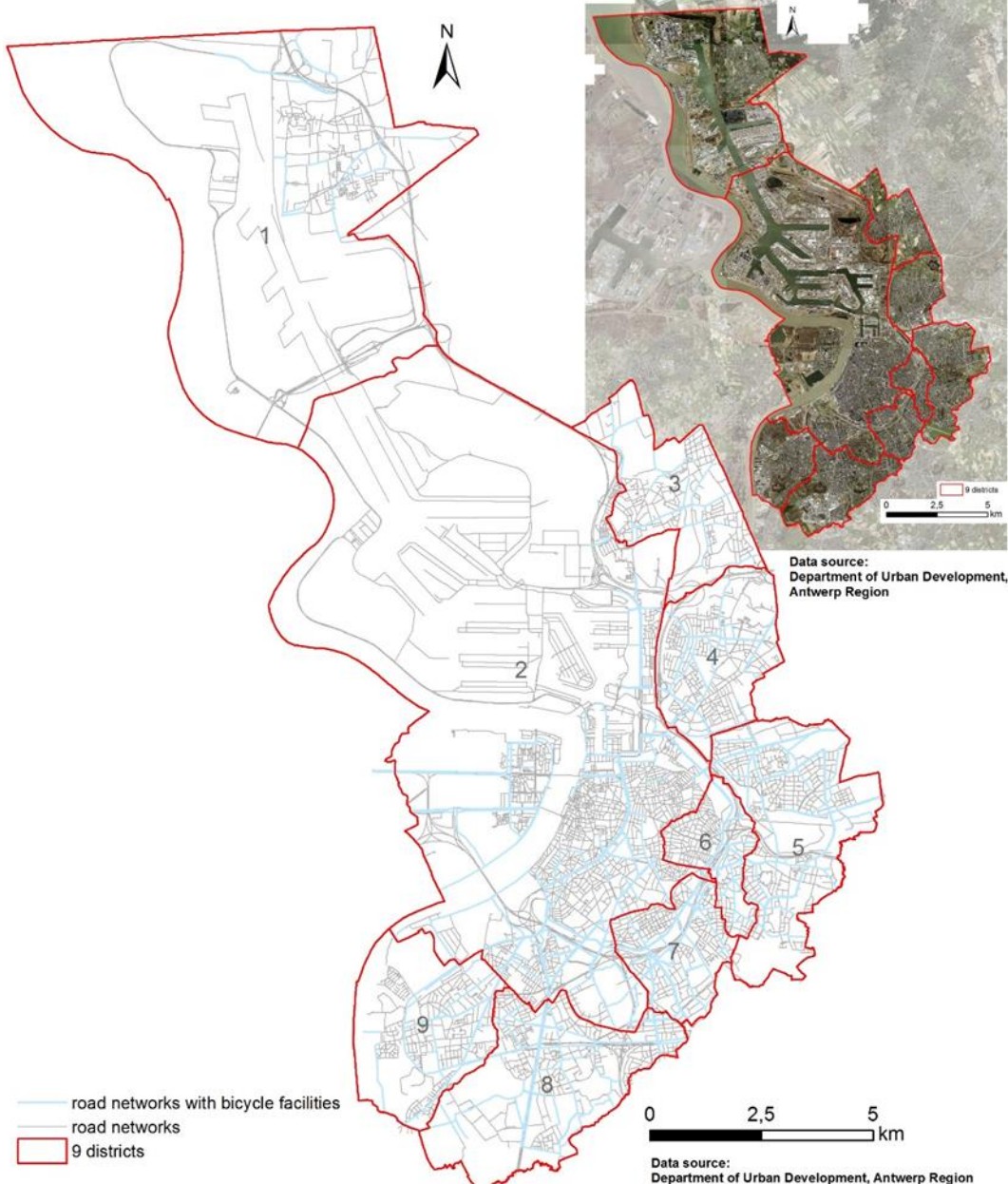

**Figure 5.** City of Antwerp (CA), with nine districts, has approximately 520,504 residents. In the CA, road networks having bicycle infrastructure are presented in blue.

In addition, in the city of Antwerp, the average family owns 2.2 bicycles and 36.5% of the traffic modal share is from cycling traffic [85]. The current amount may be higher because within each studied year, the total amount of BMV crashes has steadily grown [86].

## 5. Data Attributes and Sources

Based on the GIS road networks and reported bicycle crashes from the police, the study developed a database. Figure 2 of Section 3 demonstrates the procedure of data collection. First, stage 0 conducted a complete literature review on the influencing factors. Bicycle crashes were then geocoded on the road network of the GIS (stage 1) and their road environment characteristics were determined from police reports, field surveys and aerial photos (stage 2). Finally, the database with influencing risk factors, information attributes and bicycle crashes was applied for the estimation of bicycle–motorized vehicle crash risks in the city of Antwerp. All information in the database was collected from (1) Police

Department, CA; (2) Flemish Traffic Center; (3) Department of Transportation, CA; (4) Department of Urban Development, CA; (5) Port Authority, CA; and (6) field surveys conducted specifically for the study (see Table 1, column 3).

The following attributes were contained as a crash event's information: date, time, location, the weather, vehicle type, crash severity, crash type, lighting, road surface condition of the road and risk influencing factors (Table 1, column 1). Table 1, column 2, fully describes these risk factors and categorises them into three classifications: road environments, road traffic controls and road traffic facilities. Although the database did not contain information about liability, the scene of a bicycle crash was easier to reconstruct by using crash scene photos, standard crash report forms, official police reports and crash site figures depicted by trained police personnel. Bicycle crashes involving at least one motorised vehicle were considered in this study, resulting in 4162 reported crashes in this spatio-temporal database from January 2014 to December 2018. However, due to the invalidation of some crash locations, 42 reports were excluded. Finally, 4120 crashes (stage 1) and 1415 arterial crashes (stage 2) were the objects of analysis in this study.

## 5.1. Analytical Environments

By exchanging file formats and combining three analysis software programs, the spatio-temporal approaches, introduced in Section 3, were built. ArcGIS 10.6.1 performed all spatial analysis, while Limdep 11 and SPSS 24 (S. IBM: Armonk, NY, USA, 2016) accomplished all temporal and statistical analysis. Manual input of police reports and address mapping successfully geocoded 99% of the bicycle crashes. By using the adaptive bandwidth [87], ArcGIS 10.6.1 performed kernel density estimations, obtaining high-density areas of BMV crashes.

**Table 1.** The variables used in the modelling equations, and the influencing factors used in the spatial and temporal approaches.

| Variables and Influencing Factors | Defined Categories | Data Source | Relevant Literature |
|---|---|---|---|
| Road networks | (ref. number of road forks) | The road network of ArcGIS from (3) and (4) or from Google Map | [44] |
| BMV crash at location $i$ | $i$ = 1–4120, 1415 observed samples were on urban arterials | Crash site figures from (1) | [20,71] |
| BMV crash frequency at location $i$ | 1~45 | Original datasets from crash site figures (1), GIS Analysis through KDE | [88–91] |
| Daily bicycle flows (ADB) $F_i$ | Original counts = 44–7633 log-transformed ADB = 1.64–3.88 | Data reports from (2); data of traffic sensors from (2) and (6); data mainly from (3) | [53,71,72] |
| Daily traffic flows (ADT) $V_i$, expressed in passenger car equivalent (PCE) | Original counts = 7003~69,982 log-transformed ADT = 3.85–4.85 | Data reports from (2); data of traffic sensors from (2) and (6); data mainly from (3) | [53,71,91–93] |
| Road categories | 0 = others, 1 = rural roads, 2 = urban secondary roads, 3 = urban arterials | Annual aerial photographs from (5); ArcGIS road networks from (3) and (4); Google Map | [4,44,51,53,56] |
| **Road Environments** | **Defined Categories** | **Data Source** | **Relevant Literature** |
| Morning peak hour volume (M-PHV) (PCU) | Log (M-PHV) = 2.59–3.80 Original counts = 396–6381 | Data reports from (2); data of traffic sensors from (2) and (6); data mainly from (3) | [52,76,94] |
| Afternoon peak hour volume (A-PHV) (PCU) | Log (A-PHV) = 2.45–3.71 Original counts = 284–5116 | Data reports from (2); data of traffic sensors from (2) and (6); data mainly from (3) | [52,76,94] |
| Month (seasonal patterns) | 1–12 = January–December | Brief description of the crash from (1) | [24] |
| Day (daily patterns) | 1–7 = Monday–Sunday | Brief description of the crash from (1) | [24,51] |
| Weekend (weekly patterns) | 0 = weekday, 1 = weekend | Brief description of the crash from (1) | [24,51] |
| Hour (hourly patterns) (h) | 1~24 | Brief description of the crash from (1) | [24] |

**Table 1.** *Cont.*

| Road Environments | Defined Categories | Data Source | Relevant Literature |
|---|---|---|---|
| Crash severity index (CSI) | 0 = $A_4$ property damage, 1 = $A_3$ slight injury, 2 = $A_2$ severe injury, 3 = $A_1$ fatality | Data from (1) and hospital records | [4,21,53,59,91,93,94] |
| Light conditions | 0 = daytime, 1 = night-time | Brief description of the crash from (1) | [4,44,56,63,95,96] |
| Number of lanes (unidirectional) | 0–6 | Aerial photos from (4) and (5); crash scene photos, brief description of the crash, crash site figures from (1); | [4,92,93,97,98] |
| Length of segments (m) | Log(LoS) = 1.05–2.72 Original length = 11.17–521.79 m | Road networks of ArcGIS from (3) and (4); crash site figures from (1); Annual traffic engineering facilities of AutoCAD measurements, from (4) | [12,20,99] |
| Area of junctions ($m^2$) | Log(AoJ) = 0, 1.50–4.26, Original counts = 31.79–18,177.19 $m^2$ | Road networks of ArcGIS from (3) and (4); crash site figures from (1); Annual traffic engineering facilities of AutoCAD measurements, from (4) | [20,100] |

| Road Engineering Facilities | Defined Categories | Data Source | Relevant Literature |
|---|---|---|---|
| Road section/Intersection | 0 = intersection, 1 = road section | ArcGIS road networks from (4); Google Map; Auto Cad Map | [4,30,59,90,94] |
| Lighting systems | 0 = at daytime, natural light; 1 = at night-time, natural light; 2 = at night-time, with lighting; 3 = at night-time, without lighting | Brief description of the crash from (1) | [4,44,51,56,63,95,96] |
| Residential area | 0 = not adjacent residential areas, otherwise = 1 | ArcGIS residential zones from (3) and (4) | [12,41,51] |
| Major road | 1 = intersect with another major road, otherwise = 0 | ArcGIS road networks from (2), (3) and (4) | [12,45,51] |
| Secondary road | 1 = intersect with a secondary road, otherwise = 0 | ArcGIS road networks from (2), (3) and (4) | [12,24,31,45,51] |
| Collector road | 1 = intersect with a collector road, otherwise = 0 | ArcGIS road networks from (2), (3) and (4) | [12,24,45] |
| Central business district (CBD) | 1 = adjacent CBDs, otherwise = 0 | Annual ArcGIS CBD zones from (4) and (5) | [7,71] |
| One-way bicycle path | 1 = with a one-way bicycle path, otherwise = 0 | Original data from (3) and (4) and field investigation (6) | [7,12] |
| Two-way bicycle paths | 1 = with two-way bicycle paths, otherwise = 0 | Original data from (3) and (4) and field investigation (6) | [7,12] |
| Two-way turns into a one-way bicycle path | 1 = between a one- and two-way bicycle paths, otherwise = 0 | Original data from (3) and (4) and field investigation (6) | [7,12] |
| Distance from the school (m) | 0 = 1–200 m from the school, 1 = 201–400 m, 2 = 401–600 m, 3 = more than 600 m | The location of crashes and schools from (1) and (4) respectively; distance measured by AutoCAD, original data from (2), (3) and (4); | [41,90] |
| Tram tracks | 1 = with tram tracks, otherwise = 0 | The location of tram tracks from (4); | [7,23,46,101] |
| Distance from the bus stop (m) | 0 = 1–200 m from the bus stop, 1 = 201–400 m, 2 = 401–600 m, 3 = more than 600 m | The location of crashes and bus stops from (1) and (4) respectively; distance measured by AutoCAD, original data from (2), (3) and (4); | [31,90,101] |
| Bus routes | 1 = passing through bus routes, otherwise = 0 | Crash site figures from (1); ArcGIS road networks from (3) and (4) | [31,90,101] |
| Main cycling routes | 1 = within main cycling routes, otherwise = 0 | ArcGIS road networks from (3) and (4); field investigation (6) | [101] |
| Lane marking | 0 = no lane marking, 1 = lane marking, 2 = lane marking with directional arrows | Crash site figures, brief description of the crash, the annual dataset of facilities from (2), and crash scene photos from (1) | [21,91,102–104] |
| Numbers of lanes (uni-directional) | 0–6 | Annual datasets of facilities from (2); crash scene photos, brief description of the crash, crash site figures from (1) | [4,92,93,97,98] |

**Table 1.** *Cont.*

| Road Traffic Controls | Defined Categories | Data Source | Relevant Literature |
|---|---|---|---|
| Speed limits (km/h) | 20~90<br>20, 30, 50, 70, 90 | ArcGIS road networks from (3) and (4); annual datasets of facilities from (2) | [4,44,53,56,59,71,97,103] |
| Signalised facilities | 0 = no, 1 = flashing amber signals,<br>2 = traffic signals,<br>3 = traffic signals prioritise cyclists and pedestrians | Data from (2) and (3) and crash site figures from (1) | [30,46,53,71,93,100,102,105] |
| Signal cycle lengths (s) | 0 = 0~60, 1 = 61~120,<br>2 = >120 s | Data from (2) | [91,101,106,107] |
| Zone 20/30<br>(residential zone) | 0 = non-traffic-calming zone,<br>1 = within dynamic traffic-calming zone, 2 = within zone 30 km/h,<br>3 = within zone 20 km/h, | Data from (2); annual ArcGIS traffic calming zones from (4) and (3) | [39,41] |
| One-way road | 1 = one-way road,<br>otherwise = 0 | Annual aerial photographs from (5); ArcGIS road networks from (3) and (4); | [12,20,51] |
| Turning movement of motorists | 0 = straight ahead, 1 = left turns,<br>2 = right-turning | Crash site figures and brief description of the crash from (1), and sensor data from (2) | [4,93,100] |
| Turning movement of cyclists | 0 = straight, 1 = left-turning,<br>2 = right turning | Crash site figures and brief description of the crash from (1), and sensor data from (2) | [4,93,103] |

Data source: (1) refers to Police Department,CA; (2) refers to Flemish Traffic Center, Belgium; (3) refers to Department of Transportation (DOT), CA; (4) refers to Department of Urban Development (DUD), CA; (5) refers to Port Authority, CA; and (6) refers to field surveys conducted specifically for this research. In stage 1: only 15 variables were available for the spatial analysis in ArcGIS 10.3 (4120 observed BMV crashes), which were crash type, crash time, severity index (CSI), road section/intersection, lighting systems, residential area, urban arterials, minor roads, collector roads, CBD, bicycle paths, tram tracks, bus routes, speed limits, signalized facilities and zone 20/30. No spatial correlations are found among these variables.

## 5.2. Influencing Risk Factors

This study's purpose was to understand the spatial and temporal dynamics of road environments and engineering facilities. Other influencing risk factors, like the gender or age, could also provide important information for BMV crash prevention or intervention. However, it was not permissible to obtain data on human factors that might lead to identification of individuals in this research due to the privacy concerns [24]. Table 1 not only shows all of the risk factors, data sources, and detailed descriptions utilised in this study, but also mentions the relevant studies of these risk factors: (1) City of Antwerp (CA) had tram tracks and icy roads; (2) bicycle and traffic exposure data were acquired from Flemish Traffic Center (FTC) or from field surveys with the traffic cameras; (3) the study considered two traffic-calming measures: residential zones (30 and 20 km/h), and dynamic traffic-calming zones (car-free zones) during certain hours; and (4) the study also contained traffic signals, and turning movement of bicycles and motorised vehicles, since these were likely to be influencing factors leading to BMV crashes.

## 6. Results

This section concisely describes the spatial and temporal pattern of bicycle crashes. For the spatial scale, bicycle crashes tended to aggregate on arterial roads (see Figure 6c–f). A total of 38.42% and 35.68% of arterial crashes were caused by rear-end conflicts and side crashes with overtaking manoeuvres, respectively. Around 10% of BMV crashes were situated in the zone hotspots (see Figure 7). Within these zone hotspots, 78.52% of crashes were associated with urban arterials and 50.26% with road junctions. Similarly, throughout the city of Antwerp (CA), 21.70% of arterial crashes were in the red zones (zone hotspots), whereas only 3.01% of secondary road crashes were in these red zones. Arterial crashes were unevenly distributed and were disproportionately related to the length of the network. Figure 7 shows approximately 33.61% of bicycle crashes were concentrated in 28.79% of the total network length (i.e., the network length of arterials).

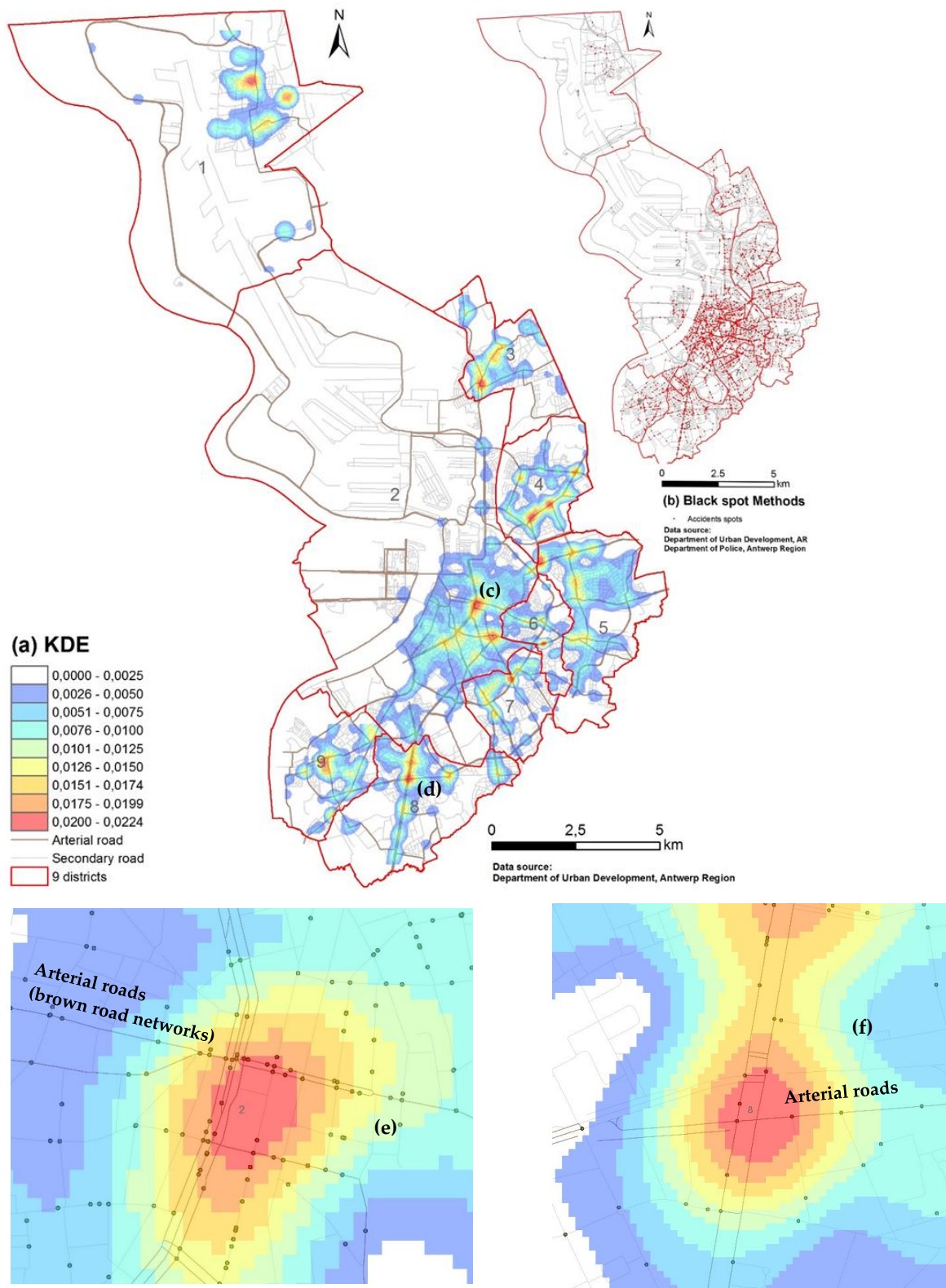

**Figure 6.** Spatial approaches: (**a**) the result of the kernel density estimation (KDE). The network of arterial roads is shown in brown, secondary roads are shown in grey, and the high-risk areas of BMV crashes are shown in red (zones); (**b**) the spatial distribution of bicycle crashes in the CA (black spot approach); (**e**) and (**f**) the partial enlarged figures of the spatial approach (KDE) from Figure 6 (**c**) and (**d**) respectively: bicycle crashes were highly aggregated on arterial roads. BMV crashes are displayed in black (points) and arterial roads are displayed in brown (lines).

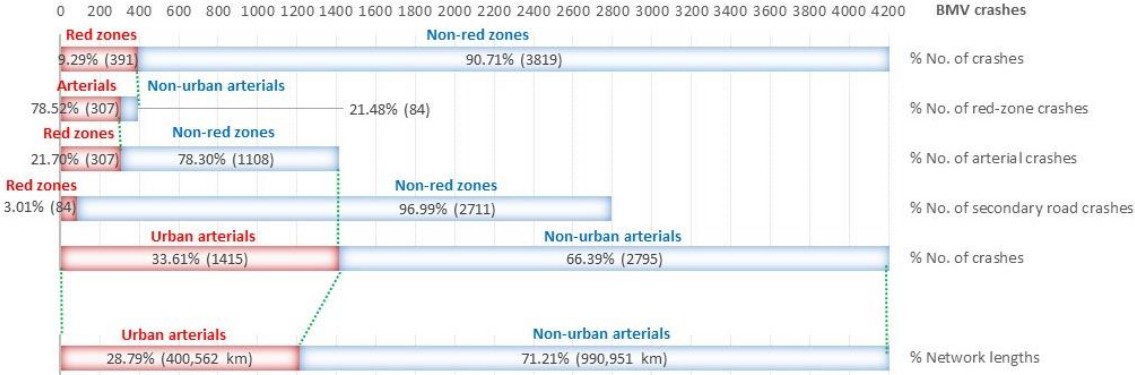

**Figure 7.** Spatial distribution of bicycle–motorized vehicle crashes.

For the temporal scale, arterial crashes were more likely to be clustered in summer (28.03%), on weekdays (82.99%) and during peak hours (46.41%, from 7 to 9 am, and from 4 to 6 pm). Descriptive statistics demonstrate that crash frequencies matched the Poisson distribution by following independent, scarce, and random events that occurred within the studied year, and were more consistent with the NB distribution because of their over-dispersed, non-negative features (Table 2). The frequency's expected value was 5.16 but its variance was 303.78, unequal to the mean value, which preliminarily proved the NB modelling was more suitable for the explanation of influencing factors in the study. To understand crash patterns on arterial roads, Tables 2 and 3 categorised influencing factors into continuous and ordinal/nominal variables. Since some paired influencing factors were highly correlated (the correlation coefficient ≥ 0.7), each paired dependent variable was first examined for their explanatory ability regarding BMV crash modelling, and the ones with a less favourable explanation (e.g., light condition "night-time" was replaced with "night-time with lighting" and "night-time without lighting") were then excluded from the model.

**Table 2.** Influencing factors with continuous variables and with statistical significance.

| Contributing Factors | Mean | Std Dev | Minimum | Maximum |
|---|---|---|---|---|
| **Dependent Variables** | | | | |
| Crash (4120 observations) | 3.66 | 16.94 | 1 | 45 |
| Arterial crash | 5.16 | 17.42 | 1 | 45 |
| **Contributing Factors** | | | | |
| Log(average daily bicycle flows (ADB)) | 2.59 | 1.52 | 1.64 | 3.88 |
| Log(morning peak hour volume (M-PHV)) | 3.63 | 0.78 | 2.59 | 3.86 |
| Log(morning peak hour volume (M-PHV)) | 3.60 | 0.64 | 2.45 | 3.71 |
| Numbers of lanes (unidirectional) | 2.92 | 1.43 | 0 | 6 |
| Log(length of segments (LoS)) | 2.05 | 2.21 | 1.05 | 2.72 |
| Log(area of junctions (AoJ)) | 2.99 | 3.11 | 1.50 | 4.26 |
| Speed limit $X_{i3}$ (km/h) | 52.48 | 9.226 | 20 | 90 |

Number of observations = 1415.

**Table 3.** Influencing factors with ordinal/nominal categories and with statistical significance.

| Category = 0–3 (See Table 1) | The Category and Number (%) of BMV Crash Occurrences | | | |
|---|---|---|---|---|
| | **0** | **1** | **2** | **3** |
| Historical crash severity (CSI) | 272 (19.2) | 1049 (74.1) | 87 (6.1) | 7 (0.5) |
| Daytime or night-time | 1099 (77.7) | 316 (22.3) | | |
| Intersection/Road section | 744 (52.6) | 671 (47.4) | | |
| Lighting systems | 906 (64.0) | 14 (1.0) | 199 (14.1) | 296 (20.9) |
| Residential area | 1264 (89.3) | 151 (10.7) | | |
| Intersect with a secondary road | 1294 (91.4) | 121 (8.6) | | |
| Central business district (CBD) | 1099 (77.7) | 316 (22.3) | | |
| Two-way bicycle paths | 1328 (93.9) | 87 (6.1) | | |
| Two-way turns into a one-way bicycle path | 1372 (97.0) | 43 (3.0) | | |
| Tram track | 1004 (70.0) | 411 (29.0) | | |
| Distance from the bus stop (m) | 601 (42.5) | 327 (23.1) | 275 (19.4) | 212 (15.0) |
| Bus routes | 670 (47.3) | 745 (52.7) | | |
| Main cycling routes | 975 (68.9) | 440 (31.1) | | |
| Lane marking | 830 (58.7) | 46 (31.5) | 139 (9.8) | |
| Signalised facilities | 1298 (91.7) | 117 (8.3) | | |
| Signal cycle lengths (s) | 549 (42.0) | 424 (30.0) | 397 (28.1) | |
| Manoeuvre of motorists | 710 (50.2) | 210 (14.8) | 945 (35.0) | |

Number of observations = 1415.

After running the estimation procedure in stage 2, the result of the likelihood-ratio test for the estimation of dispersion parameter ($\alpha$) shows the significance was 0.5363 with a *p*-value = 0.0001. This meant that crash data were over-dispersed and NB modelling at this statistical significance was superior to and thus replaced Poisson modelling. This result reconfirmed that the NB modelling was more suitable for evaluating bicycle crashes in the city of Antwerp. By removing certain variables (e.g., light conditions) to solve correlations, 32 independent influencing factors remained in the final risk model. To identify significant influencing risk factors, NB modelling was first performed in the temporal workflow. Second, to enhance the modelling performance, the same NB modelling was performed using only influencing factors with significant levels. Figure 8 shows the estimated coefficient and their *p*-values for these risk factors. The coefficient of these factors indicated the comparative risk level of the overall NB modelling. A positive coefficient shows that *Xi* was a possible aggravating factor related to bicycle crashes, while a negative value suggested a mitigating factor, corresponding to Equation (3) (see Methodology). For example, a higher risk of BMV crashes was found in locations during the morning or afternoon peak hours. Oppositely, a lower risk was found in locations with main cycling routes or with signalised facilities. The results show that more than two thirds of the influencing risk factors were significant in the temporal model (Figure 8a–c), thus giving a suitable classification under the model.

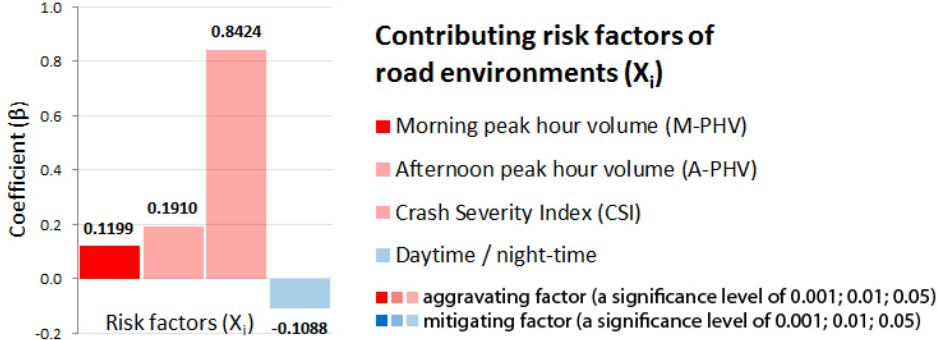

(**a**) contributing risk factors of road environments (only significant factors are listed).

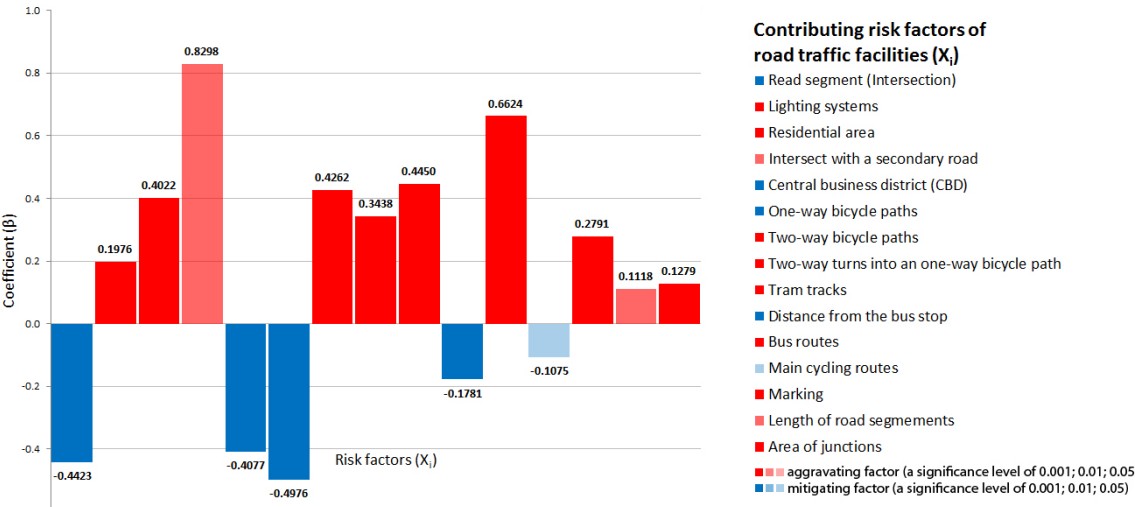

(**b**) contributing risk factors of road engineering facilities (only significant factors are listed).

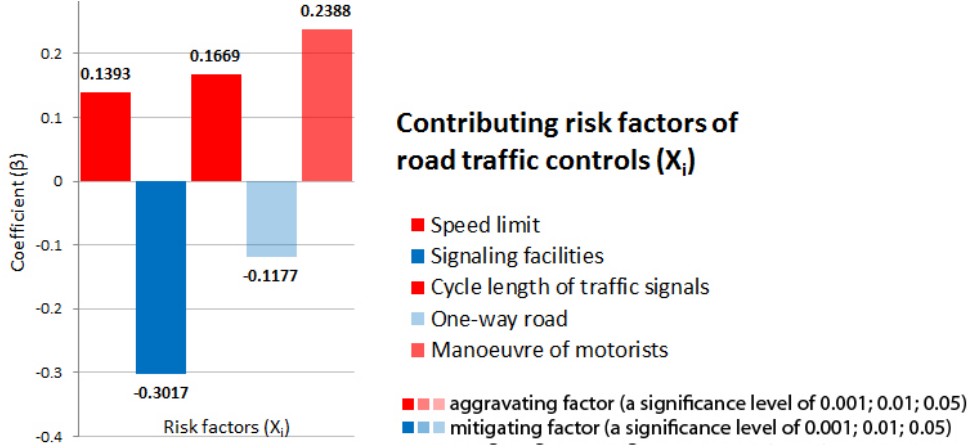

(**c**) contributing risk factors of road traffic controls (only significant factors are listed).

**Figure 8.** The result of temporal approaches (stage 2).

## 7. Discussion

### 7.1. Spatial Dimension

In stage 1, the 4120 reported crashes appeared at 1125 different locations of the CA. Within the studied areas, the crashes were strongly aggregated, as shown in Figure 6. During the entire survey period, the areas with the highest bicycle crash density occurred in districts 2, 4, 5, 7 and 8. These areas are located in the city centre, enclosed by the ring road R1 from the east, south and west, and have a relatively large number of bicycle facilities (see Figure 5) and urban road networks (see the top right corner of Figure 6). Arterial roads were identified as having the highest density of bicycle crashes in the city of Antwerp. A total of 85% of all arterial crashes were situated in these five (2, 4, 5, 7 and 8) districts. Descriptive statistics at this stage indicate that other crash features (e.g., road conditions, road engineering facilities, road traffic controls) in these districts did not significantly differ from the rest of the city of Antwerp.

The spatial aggregation of bicycle crashes was consistent with the study by Lovelace et al. [63] and Loidl et al. [24] who found significant spatial aggregations of bicycle crashes on the macro (regional) scale. The former also used KDE for the spatial approach. However, through a temporal approach, it becomes more essential for this case study to pay further attention to arterial crashes on the micro (locational) scale.

### 7.2. Temporal Dimension

Stage 2 focused on the arterial crash analysis. The 1415 reported arterial crashes were located at 274 different locations. The temporal pattern of BMV crashes was revealed by influencing risk factors (e.g., the relation of traffic flows and road environments [108], dynamic road traffic controls [46,53,93]). The results are discussed based on the existing bicycle crash literature.

#### 7.2.1. Road Environments

The results reveal that traffic exposure may provide a significant contribution to the risk of bicycle crashes on urban arterials. During peak hours in the morning and afternoon, increased traffic may result in an increase in bicycle crashes. However, the degree of their influence is not huge, expressed by the coefficients of 0.1199 and 0.1910, respectively. This may be due to the fact that road users typically slow down under higher traffic flows during peak periods, thereby reducing the impact of urban traffic flows. This result is also confirmed by previous studies [3,71,109,110]. However, the results reveal that bicycle flows were not significant in aggravating the risk of bicycle crashes in the city of Antwerp. This may be attributable to the concept of "safety in numbers" [36,111] and is especially suitable for reported crashes (major and fatal injuries) [110,111]. Such a result may be caused by the effect of "risk compensation". A possible explanation for this phenomenon is that motorists may drive more carefully when they see groups of cyclists, implying that group cycling may change driving behaviour [7,34].

The study has also demonstrated that injury severity was positively related to the risk of BMV crashes, expressed by the coefficient of 0.8424. This means that an urban arterial section with higher crash frequencies may be considered more dangerous because it also had more serious bicycle injuries [20,21,112]. In addition, while bicycle crash risks at arterials were rarely affected by changes in the season, day or hour, darkness was a common influencing factor for reducing bicycle crash risks, which is consistent with previous findings: night-time may contribute to a decline in cycling, especially during wintertime [113], and sunshine is identified as an influence on bicycle ridership [114]. Although the risk of bicycle crashes was much higher in daylight than in darkness, our study has still indicated one out of five severe crashes were related to darkness, implying that improved lighting conditions or wearing visible clothing may reduce the value of the crash severity index (CSI) [69,115], and then may indirectly lower the risk of BMV crashes.

### 7.2.2. Road Engineering Facilities

Road engineering facilities have a significant impact on the risk of BMV crashes. This risk may be attributed to the geometric design of the road environment. Road sections (i.e., the absence of intersections) may significantly lower the risk of bicycle crashes, as expressed by the coefficient of −0.4423 (*p*-value < 0.01). This result concurs with recent studies that show the majority of BMV crashes happened at road junctions [7,12,31,116]. The absence of lighting systems at night influenced the risk of BMV crashes, as shown by the coefficient of 0.1976 (*p*-value < 0.05). This is consistent with the existing literature, indicating that inadequate road lighting is an important risk factor [69,115] and sufficient lighting may reduce bicycle crashes by 58% [117]; road lighting may improve the visibility of cyclists, thus reducing the risk of BMV crashes.

In addition, road categories and land use may also influence crash risks, as observed by Kaplan and Giacomo Prato [35], as well as Vandenbulcke et al. [7]. Regarding road categories, due to mixed traffic with heterogeneous vehicular velocities, urban arterials intersecting with secondary roads and residential areas may increase cycling risk, as shown by the coefficients 0.8298 and 0.4022, respectively. Secondary roads were twice as risky as residential areas because of more complex traffic situations. Implementing protected or divided bicycle facilities may greatly reduce the crash risk. Residential areas connected to arterial roads were related to increased bicycle crash risks. This may be because in narrow neighbourhood streets, the conflict between cyclists and motorists is more likely and the majority of neighbourhood streets lack separate bicycle paths. This conclusion is consistent with findings by Chen et al. (2018) [118]. Therefore, the implementation of one-way roads with bicycle paths may be an effective countermeasure to reduce risk. Second, crash risks may also be influenced by land use features [67]. In Antwerp, many roads within the central business areas were constructed earlier than the average urban road and have heavy traffic during rush hours. However, arterial roads located in the central business district (CBD) had lower crash risks. This may represent the effectiveness of municipal efforts to improve bicycle safety. For example, most arterial roads located in the CBD have been monitored with strict traffic law enforcement, thus enhancing bicycle safety [71]. Another possible explanation is that areas with dense road networks are mostly CBDs where bicycle facilities are well-designed with separated lanes, thus the conflicts between cyclists and motorists do not increase with road density [118].

The crash risk may also be influenced by lane types. In the CA, most lanes on arterial roads were correlated to the heterogeneity of traffic speed and were riskier than lanes with only fast or slow transport modes. The results demonstrate that tram routes and bus routes on the mixed lane may increase crash risks (coefficients = 0.4450 and 0.6624, respectively), while one-way bicycle paths and main cycling routes seemed to decrease crash risks, as shown by the coefficients −0.4976 and −0.1075, respectively. Previous studies also confirmed that tram track or bus route crashes on mixed lanes were significantly higher than arterial roads without bicycle infrastructure [7,23,43,119], and crash risks on the road with bicycle paths, compared to the one without, are reduced by about 25% [120,121]. Additionally, the results show that a longer distance from the bus stop [116] may have a lower possibility of having crashes, as shown by the coefficient of −0.1781. Therefore, physically separated bicycle paths near bus stops or tram tracks may greatly reduce this type of BMV crash [7]. However, an exclusive lane for two-way cyclists significantly raises BMV crash risks on arterial roads (coefficient = 0.4262), mainly because motorists do not see cyclists coming from right/left directions (two directions) [43,103,122]. Moreover, when two-way dedicated bicycle lanes became one-way (or the reverse), there was an increased risk because a sudden change required more reaction time for motorists and cyclists to respond. Marking bicycle crossings with coloured pavement at intersections, or providing physically divided and protected bicycle facilities, may greatly reduce the risk of BMV crashes, thereby improving road safety.

Illegal overtaking manoeuvres [123] have also been seen as a contributory factor for the risk of BMV crashes, as shown by the coefficient 0.2791. For example, on arterial roads, black spots were situated where there are overtaking-prohibited lanes (with marked lines). The results indicate that

72% of the conflicts at these locations were inappropriate lateral collisions, while few were frontal and rear collisions, emphasising the importance of maintaining adequate lateral clearance between bicycles and motorised vehicles (see References [98,124,125]). Physical lane boundaries or divisional facilities (e.g., channelization) may prevent vehicles from lane changes, effectively reducing BMV crash risks on arterial roads (Wang et al., 2019) [20]. Proper mitigating countermeasures for arterial roads (e.g., the expansion of the sidewalk for the purpose of cycling, the reallocation of bicycle paths on the sidewalk, the implementation of segregated bicycle paths or the narrowed width of traffic lanes to prevent motorised vehicles from inappropriate overtaking manoeuvres) may significantly lessen the risk of BMV crashes.

　　The dimension of road engineering facilities [98,123] may also have a significant effect on the risk of BMV crashes. The risk may be attributed to the increased size of arterial road junctions. In the CA, large-scale junctions often face far more complex traffic situations, such as high volumes of traffic, complex traffic compositions and speed differences between motorised vehicles and bicycles, thereby raising the risk of BMV crashes on arterial roads. Additionally, a road section with a longer length may raise the risk of BMV crashes, as shown by the coefficient 0.1279. In other words, increased length of road sections may result in increased activities of cycling, leading to a raised risk of cycling-related crashes and injuries [12,20,126].

### 7.2.3. Road Traffic Controls

　　Finally, the risk of BMV crashes may be associated with road traffic controls. This study has demonstrated that roads with higher speed limits may raise the risk of bicycle crashes. This study has further shown that roughly 75% of bicycle crashes took place on urban arterials with speed limits ranging from 50 to 70 km/h. Therefore, at some crash-prone locations, vehicles on arterial roads might be advised to limit their speed to below 50 km/h [4], and cyclists are advised to have a minimum sufficient sight distance of 38 meters [127,128]. In addition, when the speed limit of arterial roads exceeds 50 km/h, shoulder curbs or curb lanes should separate higher traffic flows from cyclists (of note, the Danish Cycling Embassy recommends a threshold of 60 km/h) [129].

　　The existence of signalling facilities may reduce the risk of BMV crashes. These facilities reduce the speed of vehicles, significantly decreasing the risk of conflicts between bicycle and motorised vehicles [130], as seen with a coefficient of −0.3017. The results are similar to those of Sweden [105] and Canada [30], suggesting that the presence of bicycle lanes, in conjunction with traffic signals, may notably reduce the risk of bicycle injuries. From the classification of these signals (Table 1, column 3), traffic signals prioritising pedestrians and bicyclists are advised as a measure to prevent BMV conflicts and improve road safety.

　　An increased cycle length of traffic signals (i.e., more than 120s to complete a cycle of green, amber, and red indications for both bicycle and motorised vehicle phases together) may result in a high risk of BMV crashes, as seen with the coefficient of 0.1669. The longer time of green indications may induce aggressive driving behaviour (e.g., violation of road markings, failure to make way, inappropriate lane-changing, and the increase of travelling speed [20]). On the other hand, the increased cycle length may entail a longer duration of red indications for cyclists. Bicycle queues may enlarge, occupying the whole lane (i.e., "spread effects" [107]), creating more conflicts between bicycles and motorised vehicles. Therefore, it is recommended that bicycle signal phases be added [131] or to appropriately shorten the cycle length of traffic signals to prevent erratic driving behaviour and potential conflicts, thereby lessening the risk of BMV crashes and injuries [106].

　　One-way roads [51] were correlated with a decreased risk of bicycle crashes (coefficient = −0.1177). One-way roads are expected to be safer than two-way roads because traffic situations are less complex, enabling motorists to more easily notice cyclists [51,132]. One-way arterial roads may also result in fewer crashes involving bicyclists because there are fewer turning movements. The results also demonstrated that motorist manoeuvres [133] induce a high risk of bicycle crashes (coefficient = 0.2388). More than half of the crashes (55.6%) occurred when cyclists were riding in a straight line and drivers

were turning, similar to prior data [4]. One probable explanation is that motorists turning right may pay more attention to motorised vehicles or bicycles coming from the left, thus failing to see cyclists from the right until it is too late to react. However, the crash risk may still be decreased with proper interventions at arterial junctions with high turning manoeuvres (e.g., adopting bicycle (green) signal phases [116], implementing coloured bicycle crossings, eliminating junction bus stops or using junction refuges), which may greatly reduce crashes with left-turning and right-turning vehicles [134].

## 8. Conclusions

### 8.1. Meaning of the Two-Stage Workflow

A two-stage workflow focuses on bicycle and motorised vehicle crashes and combines advantages from recent studies conducted in geography and transportation fields. By using a two-stage strategy to assess bicycle crash risks, a spatio-temporal workflow opens new research directions for the analysis of traffic crashes (i.e., models aiming at estimating the risk of BMV crashes from a macro scale (a region) to a micro scale (a location/road network)). Compared to conventional methods for the analysis of bicycle safety (such as a crash frequency model), the adoption of this two-stage strategy has a number of methodological advantages:

(1) In stage 1, the evaluation of bicycle crash risks had already been made possible despite inadequate bicycle traffic exposure; (2) both census/population data (stage 1) and exposure data (stage 2) were included in the models, improving the accuracy of estimates; (3) in the second stage, the detailed data collection of each crash point avoided potential errors arising from the arbitrary aggregation of point data (crash points) in the first stage, thus reducing the risk of point data aggregation; (4) the characteristics of bicycle crashes could be better understood through visual and statistical analysis from a macro- to a micro-level; (5) stage 1 avoided the expensive work of collecting counting data, thus stage 2 minimised labour-intensive and time-consuming analyses, providing conclusions about the influence of different environment conditions and road facilities; (6) in comparison with the traditional crash black-spot approach (Figure 8b), the prediction of potential crash risks could be provided for locations where bicycle crashes were unknown or underreported; (7) finally, stage 2 could resolve missing values as the scope had reduced from the regional scale to the local scale and could be achieved via field investigation and manual inputs from original reports (e.g., crash scene figures).

### 8.2. General Conclusions

In the case study, BMV crashes may be explained by a series of spatial and temporal phenomena. This study utilised the two-stage workflow, aiming to better understand "when" and "where" BMV crashes appeared from a regional (macro) scale to a locational (micro) scale. This study supports the use of the two-stage strategy because regional studies are not suitable for locational risk assessments [24]; however, locational studies may overlook the overall tendency of crashes on the regional scale. Although the study results are specifically associated with the presented case study, general conclusions may be drawn.

(1) The two-stage workflow may capture the patterns of BMV crashes in the city, thereby measures can be suggested to reduce bicycle crashes and crash risks. (2) This strategy may also be applied to other disciplines (or other cities) and makes analysing point events possible over space and time because stage 1 may be viewed as an initial step for the identification of hot-spot areas, where these risky areas may be further addressed carefully in stage 2. (3) Up to now, few studies have focused on the pattern of BMV crash risks from a macro- to a micro- scale, mainly due to a lack of available data. Through stage 1 to stage 2, the study makes the investigation of traffic and bicycle flows possible, thus understanding the influence of traffic controls, engineering facilities and road environments on BMV crashes. (4) Utilising spatio-temporal approaches to assess crash risk is more effective than utilising conventional black-spot approaches (Figure 6b). Spatio-temporal approaches enable the potential crash risk of each location in the entire region to be calculated such that the hazardous areas

can be further addressed according to their different road characteristics (e.g., road pavement materials, type of lane, and traffic volume, etc.). (5) Finally, the spatio-temporal approach incorporates the construction year of road engineering facilities [7], dynamic road traffic controls and directional traffic volumes, leading to a more precise analysis of existing environment conditions on the arterial road.

## 9. Limitations and Further Research

It should be mentioned that this study has several limitations. First, the collection of data is labour-intensive and time-consuming. As can be seen from Table 1 and its footnote, although stage 2 has included 33 influencing risk factors, stage 1 has made only 15 risk variables available for the statistical and spatial analysis (4120 observed crashes). Second, this study has excluded some influencing risk factors associated with privacy concerns (like vehicle and human-related factors). Finally, the study aimed to understand the influence of bicycle crash risks by evaluating the influencing factors of road environmental conditions, road traffic controls, and road facilities. Further studies might include more influencing risk factors for road safety analysis.

**Author Contributions:** Research concept: H.W., S.K.J.C., H.D.B., D.L., P.D.M.; Data collection and explanation: H.W., D.L.; Investigations: H.W., S.K.J.C.; Validation: H.W., S.K.J.C., H.D.B., D.L.; Writing—review and editing: H.W.; Supervision, P.D.M.

**Funding:** This research was funded by "Improving Bicycle Safety Program", the Taiwanese Ministry of Science and Technology, grant number 106-2917-I-002-005.

**Acknowledgments:** The authors gratefully express their appreciation to Antwerp City Government and local police for the crash data, the traffic data and aerial photographs. We also thank Frank Vangeel, Chief Inspector of the traffic department of the local Police of Antwerp Police, who kindly provided constructive remarks on this paper, Paul Sonenthal, and Ward Wildemeersch, who provided extensive editorial comments on this paper.

**Conflicts of Interest:** The authors declare no conflict of interest.

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
