# Peer review of "Integrating Spatial and Temporal Approaches for Explaining Bicycle Crashes in High-Risk Areas in Antwerp (Belgium)"

_sustainability, doi:10.3390/su11133746_

Round 1

Reviewer 1 Report

This is an extensive piece of work that shows an interesting way to analyse traffic on both a regional and local scale and thereby opening up the posibility to give far more pinpointed advice on both macro and micro-individual crash location scales. I believe this work will help the traffic/accident community further if it is well presented and easily understood by researchers and policymakers. My main concern is that the work will not get the attention it requires as the manuscript as it is at the moment is difficult to read. Both in chosen wording and in the way the sentences have been constructed. It will help the manuscript significantly if an English native speaker has a good look at it. 

Reviewer 2 Report

I find the paper interesting, with a good level of detail.

However, I found the description and definitions in section 3 poor. There are some clear typos (e.g. line 232), and some sentences are very awkward.  Equations are not well explained. What are readers suppose to make out of equation 4? is this the quantity plotted in fig 7? the text around equation 4 don't make sense to me.

There are clearly issues with mathematical symbols, with italics and subscripts.

The text needs a good proof-reading. The colour coding and figure labelling are confusing.
